# Suicidal behavior risks during adolescent pregnancy in a low-resource setting: A qualitative study

Christine W. Musyimi[1]*, Victoria N. Mutiso[1], Darius N. Nyamai[1], Ikenna Ebuenyi[2], David M. Ndetei[1,3]

1 Africa Mental Health Research and Training Foundation, Nairobi, Kenya, 2 Department of Psychology and Assisting Living and Learning Institute, Maynooth University, Maynooth, Ireland, 3 University of Nairobi, Nairobi, Kenya

* dmndetei@amhf.or.ke

## Abstract

**Data Availability Statement:** All relevant data are within the manuscript and its Supporting Information files.

### Background

Suicide is one of the most common causes of death among female adolescents. A greater risk is seen among adolescent mothers who become pregnant outside marriage and consider suicide as the solution to unresolved problems. We aimed to investigate the factors associated with suicidal behavior among adolescent pregnant mothers in Kenya.

### Methods

A total of 27 Focus Group Discussions (FGDs) and 8 Key Informant Interviews (KIIs) were conducted in a rural setting (Makueni County) in Kenya. The study participants consisted of formal health care workers and informal health care providers (traditional birth attendants and community health workers), adolescent and adult pregnant and post-natal (up to six weeks post-delivery) women including first-time adolescent mothers, and caregivers (husbands and/or mothers-in-law of pregnant women) and local key opinion leaders. The qualitative data was analyzed using Qualitative Solution for Research (QSR) NVivo version 10.

### Results

Five themes associated with suicidal behavior risk among adolescent mothers emerged from this study. These included: (i) poverty, (ii) intimate partner violence (IPV), (iii) family rejection, (iv) social isolation and stigma from the community, and (v) chronic physical illnesses. Low economic status was associated with hopelessness and suicidal ideation. IPV was related to drug abuse (especially alcohol) by the male partner, predisposing the adolescent mothers to suicidal ideation. Rejection by parents and isolation by peers at school; and diagnosis of a chronic illness such as HIV/AIDS were other contributing factors to suicidal behavior in adolescent mothers.

**Funding:** This study was supported through Grand Challenges Africa [#GCA/ISG/rnd1/135], a scheme of the Alliance for Accelerating Excellence in Science in Africa (AESA). AESA is an initiative of the African Academy of Sciences (AAS)'s the New Partnership for Africa's Development Planning and Coordinating Agency (NEPAD Agency). The sponsor of the study had no role in study design, data collection, data analysis, data interpretation, or writing of the report, or the decision to submit for publication. African Academy of Sciences: https://www.aasciences.africa/.

**Competing interests:** The authors have declared that no competing interests exist.

## Conclusion

Improved social relations, economic and health circumstances of adolescent mothers can lead to reduction of suicidal behaviour. Therefore, concerted efforts by stakeholders including family members, community leaders, health care workers and policy makers should explore ways of addressing IPV, economic empowerment and access to youth friendly health care centers for chronic physical illnesses. Prevention strategies should include monitoring for suicidal behavior risks during pregnancy in both community and health care settings. Additionally, utilizing lay workers in conducting dialogue discussions and early screening could address some of the risk factors and reduce pregnancy- related suicide mortality in LMICs.

## Introduction

Every year, approximately 21 million adolescents become pregnant in Low and Middle Income Countries (LMICs) [1]. In Sub-Saharan Africa (SSA), most adolescent pregnancies occur within the context of marriage driven by traditional practices, cultural norms and poverty or occur out of wedlock, and are often unintended [2]. A study conducted in Brazil revealed that suicidal behaviour is common in teenage pregnancies, with a prevalence rate of 13.3% and is associated with psychiatric disorders like anxiety and major depressive disorder [3]. Adolescent motherhood alone has far reaching negative effects to the progeny including elevated suicidal behaviour when the children reach early adulthood [4].

The transition from a 'school girl' to pregnancy state is not often a smooth process and may affect a woman's mental wellbeing resulting in suicidal behaviour (considered the most common cause of death among female adolescents) [5]. The maternal mortality risk in adolescents has been rated at 28% higher than older women [5] with suicidal attempts risk as high as 20% [6]. Socio-economic and environmental factors occurring just before a suicide attempt influences an individual's decision to end one's life and a better understanding of these factors may protect against suicide or provide insights on prevention strategies during this period [7]. In the attempt to standardize and improve accuracy reporting of maternal deaths, the World Health Organization (WHO) recommended the description of all critical information prior to death and identification of suicide as a direct cause of maternal death ('under other category') in the WHO application of International Classification of Diseases (ICD-10) to deaths during pregnancy, childbirth and puerperium [8]. This data is critical in establishing the magnitude of the problem. To effectively address suicidal behavior in pregnant adolescent mothers, it is imperative to understand the risk and develop mitigation plans especially at the family and community level.

To our knowledge, there is no study in Kenya that has qualitatively explored the suicidal behavior risks for adolescent pregnant mothers (using multiple stakeholders' views including adolescent pregnant mothers and both formal (health care workers) and informal health providers (traditional birth attendants). We aimed to investigate the suicidal behaviour risks of adolescent pregnant mothers. We believe that such a study would inform the development of suicidal risk reduction strategies with a focus on early detection at the community level. Additionally, implementation of the strategies would attenuate maternal and child health complications post birth. Most interventions designed for use in LMICs have addressed the need to reduce access to the means of suicide as a way of preventing future attempts. However,

identifying the principal causes of suicidal behaviour or persistent distress could provide an opportunity for prevention of suicide among at-risk adolescent mothers. Kapungu and colleagues have also affirmed that the general scarcity of research among pregnant and out of school adolescents require immediate attention [9].

## Methods

### Theoretical approach (Fig 1)

This study was contextualized using the Social Ecological Model (SEM) [10] which perceives that human interaction with different social and ecological factors at different levels has influence on individuals experience and behavior. These levels include individual, interpersonal, community and societal interactions. Examples of the individual suicide risk factors include minority identity, socio demographic (e.g. gender), psychological issues like high impulsivity, agitation low self-esteem and hopelessness which should be a basis for suicidality risk assessment[11]. Feeling of being left out and sense of being a burden contributes to the suicidal tendencies [12] and thus interpersonal high risk factors include family violence or conflicts, social isolation and dysfunctional relationships. In a community where adolescent pregnancy is considered an aberrant behavior, this is detrimental to the mental wellbeing and could lead to suicidal behavior. Therefore, some identified risk factors at this level include community perceptions and tolerance of violence against adolescent mothers and barriers to access of community integrated health care, while societal risk factors include policy issues with negative implications, economic depression, poverty and stigma [13].

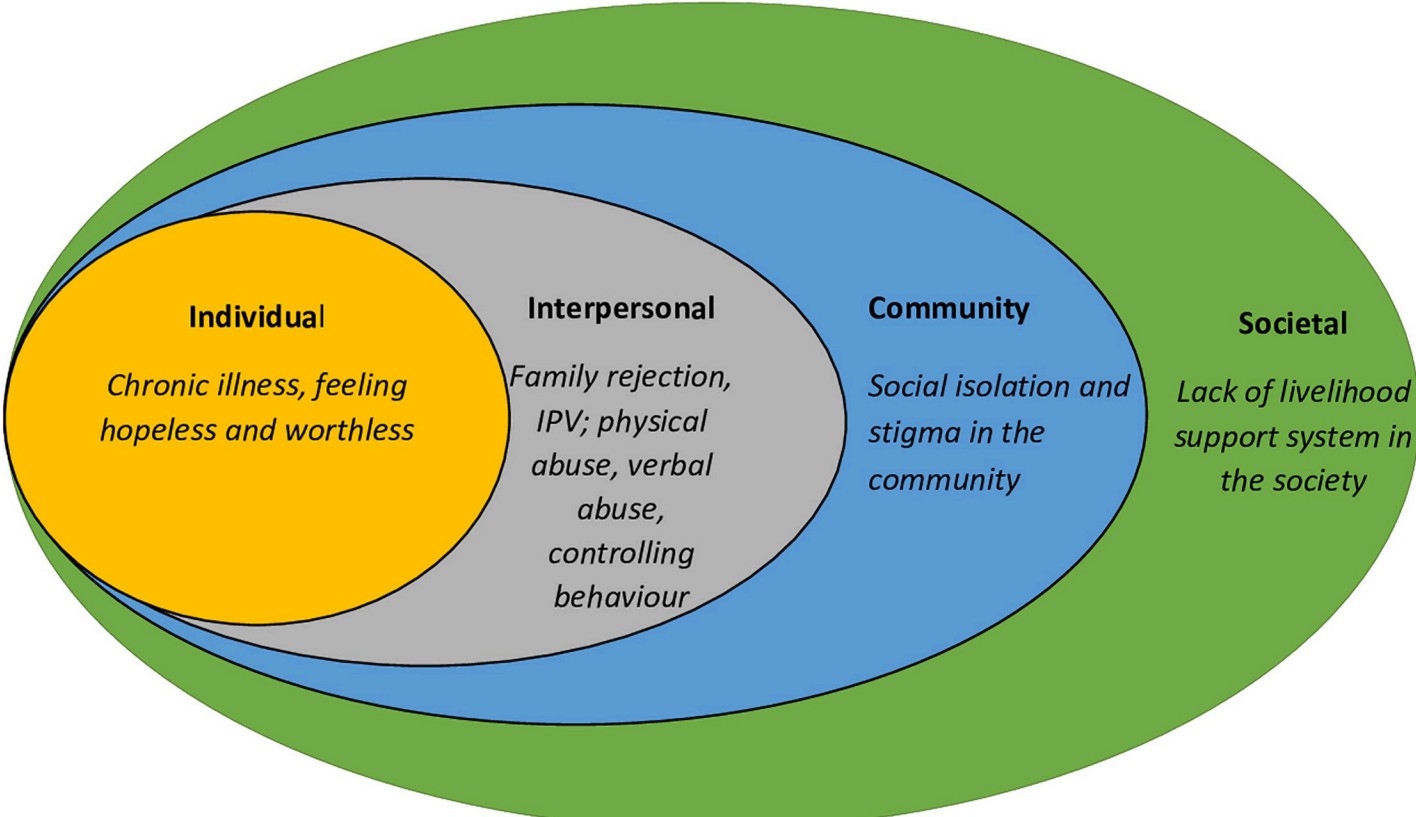

**Fig 1. An illustration of socio-ecological model highlighting suicide behaviour risks in adolescent mothers.**

## Sampling and data collection

Multiple methods were used to identify and recruit the study participants including purposive and snow balling. Initially, recruitment was done using purposive non probability sampling in Makueni County—rural setting located in lower Eastern Kenya. We used top-bottom approach to recruit participants (further described in an earlier study [14]). The Key Informant Interviews (KIIs) for community leaders and Focus Group Discussions (FGDs) for the health care workers were given priority because of their busy schedule. Community leaders were scheduled depending on their availability. The FGDs for Traditional Birth Attendants (TBAs) and the pregnant mothers preceded the care givers. This enabled us to reach out to the caregivers through snow balling because it was difficult especially to get husbands of the adolescent mothers.

In our selection criteria, all participants needed to have resided in the study area for at least six months and willing to participate in the study voluntarily. The age requirement for the adolescent participants (either pregnant or below 6 weeks post natal) ranged between 13 to 19 years. The criterion for selecting the TBAs was being 'active' with community activities. This was reflected in the monthly referrals of ante natal and post natal mothers to a primary health care facility. The public health officer in the study areas had the records of the TBA activities.

We conducted 27 Focus Group Discussions (FGDs) and 8 Key Informant Interviews (KIIs) in a quiet and private room. We targeted at least 3 FGDs for each category and at least 8 KIIs depending on the achievement of theme saturation [15]. The FGDs were conducted for; adolescent and adult pregnant mothers, and post-natal (up to six weeks post-delivery) mothers seeking care from traditional birth attendants (TBAs) and/or Health Care Workers (HCWs), Health Care Workers (HCWs) based in health care settings, TBAs, Community Health Workers (CHWs) and caregivers (husbands and/or mothers-in-law of pregnant women) [14].

The 8 KIIs were conducted with one hospital manager, two public health officers whose role is to coordinate the community health strategy, local opinion leaders (two chiefs and a village elder), post-natal woman who could not be interviewed together with the rest of the participants due to a recent still birth and one adolescent mother who was uncomfortable to participate in a FGD. The age of the adolescent pregnant mothers ranged between 15 to 19 years.

A translated contextually developed guide with questions specific to each category of participants was administered to the study participants by a multi-linguist researcher with a background in community mental health and experience in conducting interviews. A note taker with similar experience was also present to record all discussions and to document all verbal and non-verbal communication. The interviews focused on participant's views on the cause of suicidal behaviour and other mental disorders during adolescent pregnancy and the experiences of adolescent mothers in seeking care for mental disorders (especially depression and self-harm), The interviewer varied the sequence of questions as per the participants' discussion but probed for further information to seek clarity. The recordings which lasted between 30–45 minutes for KIIs and one to two hours for FGDs were transcribed, translated by a multi-linguist and merged with the notes to produce complete transcripts for each interview.

Transcripts were read by two independent researchers who came up with a set of themes while listing the key terms and explanations for each of them and specific excerpts independently added to each set of themes using Qualitative Solution for Research (QSR) NVIVO version 10 software. The coding scheme was then collectively discussed and agreed by the research team.

### Ethics statement

This study was approved by Maseno University Ethics Review Committee (MUERC) under the reference number: **MSU/DRPl/MUERC/00440/17**. Written consents and assents were obtained from all participants as well as from guardians/parents of minors included in this study. The primary authors' institutions do not have an ethics review board hence, approval was sought from a registered review Board in Kenya.

## Results

Out of the 21 adolescent mothers (age ranged from 15–19 years) interviewed during the discussions, none was employed, 60% were single, majority (60%) had attained primary school level education as the highest level of achievement while only four had completed secondary school education. More than 90% of care givers who turned up to participate in the study were mothers-in-law of the perinatal mothers. All the caregivers were unemployed and 88% had primary school as the highest level of education. For ethical reasons, it is important to note that there was no adolescent pregnancy resulting from rape.

After qualitative analysis, five main themes relating to suicidal behavior risks among adolescent mothers emerged as described by use of SEM framework. These were identified as chronic physical illnesses, intimate partner violence (IPV), family rejection, social isolation and stigma from the community and poverty. Respondents stated that exposure to extreme stressors such as IPV or rejection by the husband, mother-in-law or relatives, dropping out of school with resulting extreme financial needs increases the risk of suicidal behavior in adolescent mothers. Depressive symptoms related to the pregnancy journey and HIV diagnosis was also associated with suicidal ideation.

### Individual level factors

**Chronic physical illness.** Participants reported chronic physical illness in pregnancy as a stressor that could lead to hopelessness and suicidal behavior. The commonly mentioned physical illness was HIV/AIDs. When a pregnant adolescent was confirmed to have HIV, she would get depressed and develop suicidal ideations. These young mothers were also worried about the future of the unborn child and whenever they visited a TBA or healthcare facility, it became very challenging for the healthcare provider to effectively address their needs. This was the observation by one TBA who has been dealing with adolescent mothers for more than a decade:*"We get cases of HIV (in adolescent mothers experiencing suicidal ideations or attempts). So you find someone got pregnant without planning for it and has challenges and her brain is not settled" (37-year old single TBA)*

A primary health care worker who has interacted with adolescent mothers experiencing suicidal ideations after being diagnosed with a chronic illness during routine ante natal care check-up clinic had a similar observation:*"I can also remember another adolescent married lady who came to the clinic and when she tested HIV positive, she was very much affected, she refused to take drugs and she was not even talking to us. She was just saying "please finish up with me I want to go home". When we tried to engage her in a conversation, she was not talking but instead rejected herself so much so we realized that she was in deep thoughts and had considered committing suicide" (57 year-old nursing-officer in charge of a primary health care facility)*

### Interpersonal level

**Intimate Partner Violence (IPV).** IPV was reported in different forms including physical, verbal abuse and controlling behaviour. The male spouse would threaten to leave or make

demands to the adolescent mother to terminate the unplanned pregnancy. The persistent quarrels aggravated by substance use by the male partner and the IPV tolerance (due to taboos and stigma) in the community would expose the adolescent pregnant mothers to suicidal behavior risk. Unplanned and unwanted pregnancy, substance abuse by the male partner and mistrust issues by the spouse were the commonly stated causes of IPV.

One TBA reported the sequence of events which predisposed adolescent mothers to suicide and involved; drug abuse by male spouse, IPV, elevated stress levels for the adolescent mother, depression and suicidal thoughts. *"Young boys are taking drugs and start abusing their wives (shaking her head with frustration). The women (adolescent mothers) get stressed and have no peace. This predisposes them to mental illness such as depression and suicidal thoughts."* (67-year old married TBA)

An adolescent mother also observed: *"Some people will disagree with their husbands; especially if the husband is a drunkard. The husband comes home and arbitrarily starts to beat the wife. In the process of being battered, the young pregnant woman considers suicidal attempt so that she is not a burden to her parents and the husband"* (18-year old adolescent pregnant mother)

**Family rejection.** Family rejection and criticism was the most verbalized theme across the participants. Adolescent mothers did not relate well with parents because of unplanned pregnancy. The persistent criticism from the parents caused lack of peace of mind and brought about feelings of neglect. Due to family rejection, the pregnant adolescent contemplated suicide. Additionally, the long-term effects of the family disconnection were stated by caregivers (husbands and mothers-in-law of adolescent pregnant mothers) and included miscarriage, abortion, adverse birth outcomes such as premature birth and low birth weight.

One adolescent mother reported a difficult family experience which resulted to suicidal ideation: *"I have contemplated terminating my life, about two times using a rope. I was not relating well with my parents and I did not have peace to enable me to stay alive. At some point we were not talking to each other since they did not want to see my pregnancy* (18-year old adolescent pregnant mother)

A TBA who had been conducting counseling for young girls with depression and suicidal ideations reported: *"Last year I saw a young pregnant woman with so many problems. The husband did not want her and she was chased out of home by all family members. She told me she will commit suicide"* (36-year old widowed Traditional birth attendant (TBA)

A mother-in-law acknowledged that majority of the parents in the community reject and mistreat adolescent pregnant mothers and such rejection may have far reaching consequences.*"Most parents torture and stress their girls when they become pregnant before marriage resulting into miscarriage and abortions. This stress makes them not to eat and many end up giving birth to unhealthy babies"* (45–year old single mother-in-law of adolescent mother)

## Community level

**Social isolation by the community.** Adolescent pregnant mothers experienced mockery from their peers and community in general. This led to feelings of isolation, shame, worthlessness and rejection that resulted to suicidal ideation. One adolescent pregnant mother who had to leave school after she got pregnant explained: *"If you become pregnant while you are still in school, you might contemplate ending your life. This is because the moment your fellow schoolmates learn that you are pregnant, they keep on mocking you. In the process you develop shame and contemplate ending your life"* (19-year old adolescent pregnant mother)

Another adolescent pregnant mother also explained that there was stigma associated with pregnancy because even people in the community who supported girl education withdrew

when they discovered that the beneficiary was pregnant. Due to this stigmatizing attitude from the community, a girl could decide to commit suicide as reported by one adolescent pregnant mother.

*"In many cases they (adolescents) are adopted by churches or certain parents who go and educate them. Now when this girl gets pregnant, she feels guilty going back to her sponsors to inform them because they will get de motivated to support her. So she decides to terminate her life (18-year old married adolescent pregnant mother).*

## Societal level

**Poverty.** Lack of livelihood support in society was a major concern and a contributor to stressors which predisposed adolescent mothers to suicidal ideations. Similar to majority of the community members, adolescent pregnant mothers are not financially stable and in most cases they live with their caregivers who are economically struggling. Therefore, with no income generating activity and solely relying on their family member(s) for support or young husband who do not have a stable job, the adolescent mothers become stressed to the point of giving up. Additionally, they would at some point be viewed as burdensome. As a result, they become prone to feelings of being neglected, a situation that may lead to depression and suicidal behavior as explained by the study participants. The grinding poverty and lack of psycho-socio-economic support made it onerous to access health services.

One mother reflected on her previous experience as an adolescent pregnant mother when depression set in due to economic challenges: *"In my first pregnancy I was depressed because I got pregnant while I was still living with my parents. I was thinking too much because I did not know where to access funds to sustain me and my child after delivery as I knew my parents would withdraw since they were not really well-off to take care of both (pregnant woman and the unborn child) of us"* (35-year old married pregnant mother).

This idea was reiterated by a CHW who shared her observations on the relationship between adolescent pregnancies, depression and consequent suicidal ideation in the extremely poor resource setting. *"Sometimes you find that poverty can contribute to depression and the adolescent mother attempts suicide. This is because she got married in a family where they are struggling to get a meal in a day. The thoughts of having an additional family member in an impoverished home cause her to consider killing herself."* (43-year old married Community Health Worker (CHW).

A community leader concurred with these thoughts. *"I have shared the main factors that contribute to depression and suicide, (this) includes poverty."* (Community leader-area chief)

## Discussion

This study explored the factors associated with suicidal behavior among adolescents in a low resource and rural setting. The greatest contributor to suicidal behaviour as explained by the study participants was IPV and lack of family support. Violence during pregnancy (estimated to be 38% among low-income adolescent mothers) has been associated with a heightened risk for suicidal behavior (resulting to about 55% pregnancy-related suicides) [16], mental illnesses, miscarriage, missed antenatal visits, still birth, fetal injury, premature labour and birth and low birth weight [17]. Furthermore, women who are abused have limited ability to make decisions hence lacking control over their reproductive and sexual health [18]. They are also likely to experience repeated unwanted pregnancies due to fear resulting to an increased rate of maternal mortality, adverse health outcomes during the postnatal period and intergenerational

cycles of ill-health [19]. In SSA, violence may still persist even after the pregnancy period because of gender inequality and social acceptance of violence related to religious beliefs and patriarchal society. Since extant research has found this to be a precursor to suicidal ideas [17, 20–22], similar to the current findings, evaluating the effectiveness of interventions addressing norms and beliefs inherent in SSA may transform behaviour that support violence. A workshop that was held by the National Academy of Sciences, Engineering and Medicine [23] concluded that community-level norm change has a potential to minimize IPV in low-resource settings since these norms mediate violence and contribute to gender inequality.

Similarly, family conflict, rejection and lack of support has been considered as a trigger for suicidal behavior [24]. In this study, lack of family support was a common factor implicated in suicidal ideation and attempts among pregnant adolescent mothers. A combination of social and emotional development for adolescents, unplanned pregnancy and family rejection especially when their partners force them to get abortions or reject the child or mother-in-law's lack of support for the adolescent during pregnancy or criticisms from peers results in social isolation, feelings of guilt and hopelessness. The end-result if these triggers are not addressed is suicidal ideas, attempts and untimely deaths associated with completed suicide during pregnancy. These findings corroborate evidence from other low income settings [25] and highlights a need for parent-adolescent communication interventions [26] in order to engender family and social support. This lack of social support is further heightened by poverty which was an integral theme in this study. It is therefore not surprising that all the adolescent mothers in this study were unemployed and dependent on their families or spouse for both social and economic support. Previous studies support the causal relationship between poverty and incidence of mental illness [27].

Often times, adolescent pregnancy is unplanned and affects students from impoverished backgrounds and there may be limited opportunities for employment after dropping out of school. This predisposes the adolescent mothers to a high risk of economic insecurity and thoughts of suicide emerge. In congruence with earlier research, low-socio-economic status, unemployment and poverty [28] have been linked to suicidal behaviors.

The presence of mental disorders particularly depression was also mentioned as a risk factor for suicidal behavior in pregnant adolescents. Other physical illnesses that were major contributors included HIV, complicated by poor health-care transition [29]. Earlier evidence has demonstrated that the presence of comorbid disorders during pregnancy is often intertwined with reproductive events [30] and this increases the risk among the adolescents. Due to limited resources for mental health services in low-income settings, access to timely appropriate evidence-based care is a challenge. However, demonstrating the effectiveness of integrated mental health care or using evidence-informed models for lay workers has been shown to produce significant results [31, 32] but still requires more research to fully scale-up interventions to where they are most needed.

Findings from this study have important implications for understanding the risks for suicide attributed adolescent maternal deaths, an area that has been given little attention. Suicide is a global public health issue among adolescents and the risks during pregnancy should be addressed to save the life of the mother, the unborn child and future generation. Furthermore, prevention strategies including monitoring for suicidal behavior risks during pregnancy in health care settings and using lay workers through dialogue discussions could duly reduce the risk of pregnancy-related suicide mortality in SSA.

## Conclusion

We aimed to qualitatively investigate the suicidal behaviour risks of adolescents during the period of heightened vulnerability (pregnancy). Suicide is a rapidly growing public health

concern for adolescents and the risk is higher among adolescent pregnant mothers. Therefore, our findings add to the current understanding of prevention and management of suicidal behavior among at-risk adolescent mothers in order to attenuate maternal and child health complications in Low and Middle-Income Countries (LMICs).

Our study identified that intimate partner violence, lack of family and community support, poverty and chronic physical illness increase the risk of suicidal behaviour in adolescent mothers. Therefore, the community should explore ways of addressing IPV, economic empowerment and access to youth friendly health care for chronic physical conditions in adolescent mothers. The intervention should target all the key players including parents and the male spouse. The issue of substance use disorder as a contributing factor to IPV should be addressed at the family and community level by use of integrated approaches which are inclusive. The strategies should include monitoring for substance use disorder by the male partner and signs of suicidal behavior risks by the pregnant adolescent. This should be implemented at the primary health care facilities and also by lay workers in the community.

## Implications

Lack of elaborate support systems for the adolescent perinatal mothers makes them vulnerable to suicidal behaviour. The paucity of mental health well-being programs in the rural areas makes the situation dire. It is plausible that the government initiated youth friendly clinics can mitigate the challenges which are specific to the youth. However, the clinics are mainly found in the urban referral hospitals. We recommend that youth-tailored services should be decentralized to the rural areas and especially inclusion of mental health services for adolescents.

## Study limitation

This study was qualitative and therefore context specific. A mixed methods study could augment study findings and draw general conclusions. In terms of the study area, this research was conducted in rural settings; therefore the findings might not reflect the actual experience in urban settings. Future studies could explore suicidal behavior in the context of urban pregnant adolescent population and make informed strategic comparisons in both areas.

**Panel: Research in context.** *Systematic review*. We searched Pubmed for qualitative studies exploring suicidal behavior risks during adolescent pregnancy in Low and Middle Income Countries with the terms "suicidal behavior risks" and "adolescent pregnancy". Reference lists for identified English-language articles that met our criteria and any relevant published reviews were also scanned. We included reports published between January 1st, 2005 to October 1st, 2018. There was no report that qualitatively explored the suicidal behavior risks among adolescent pregnant mothers.

## Supporting information

**S1 Data.**
(DOC)

## Author Contributions

**Conceptualization:** Christine W. Musyimi.

**Formal analysis:** Christine W. Musyimi, Darius N. Nyamai, Ikenna Ebuenyi.

**Funding acquisition:** Christine W. Musyimi.

**Methodology:** Christine W. Musyimi, Darius N. Nyamai, Ikenna Ebuenyi.

**Project administration:** Christine W. Musyimi.

**Supervision:** Victoria N. Mutiso, David M. Ndetei.

**Writing – original draft:** Christine W. Musyimi, Darius N. Nyamai, Ikenna Ebuenyi.

**Writing – review & editing:** Christine W. Musyimi, Victoria N. Mutiso, David M. Ndetei.

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
