## [Decision Letter · Decision Letter 0]

8 May 2020

PONE-D-20-02224

SUICIDAL BEHAVIOR RISKS DURING ADOLESCENT PREGNANCY IN A LOW-RESOURCE SETTING: A QUALITATIVE STUDY

PLOS ONE

Dear Dr. Musyimi,

Thank you for submitting your manuscript to PLOS ONE. After careful consideration, we feel that it has merit but does not fully meet PLOS ONE’s publication criteria as it currently stands. Therefore, we invite you to submit a revised version of the manuscript that addresses the points raised during the review process.

We would appreciate receiving your revised manuscript by by 28th May 2020. To enhance the reproducibility of your results, we recommend that if applicable you deposit your laboratory protocols in protocols.io, where a protocol can be assigned its own identifier (DOI) such that it can be cited independently in the future. For instructions see: http://journals.plos.org/plosone/s/submission-guidelines#loc-laboratory-protocols

We look forward to receiving your revised manuscript.

Kind regards,

Russell Kabir, PhD

Academic Editor

PLOS ONE

Journal Requirements:

2. Please ensure that you refer to Figure 1 in your text as, if accepted, production will need this reference to link the reader to the figure.

Reviewers' comments:

Reviewer's Responses to Questions

**Comments to the Author**

1. Is the manuscript technically sound, and do the data support the conclusions?

Reviewer #1: Yes

Reviewer #2: Yes

2. Has the statistical analysis been performed appropriately and rigorously? 

Reviewer #1: Yes

Reviewer #2: N/A

3. Have the authors made all data underlying the findings in their manuscript fully available?

Reviewer #1: Yes

Reviewer #2: Yes

4. Is the manuscript presented in an intelligible fashion and written in standard English?

Reviewer #1: Yes

Reviewer #2: Yes

5. Review Comments to the Author

Reviewer #1: ADOLESCENT PREGNANCY IN A LOW-RESOURCE SETTING is an important issue related with suicidal behavior, however understudied. I thank authors for their important work in a country setting like Kenya which could provide insights to build the national suicide prevention strategies. Thanks.

Reviewer #2: 1. Refer to page 11 (Introduction section); Rephrase the sentence “We aimed to investigate the suicidal behaviour risks of adolescent pregnant mothers to inform the development of suicidal risk reduction for early detection at the community level and to attenuate maternal and child health complications post birth.”. It is too complex. It may be broken to two or three sentences.

2. What was the selection criteria for choosing the participants for FGDs?

3. Whether the authors also included adolescent mothers, who became pregnant after non-consensual sex / rape?

4. The authors need to mention the limitations of this study and recommendations to overcome limitations.

5. In the implications, the authors may discuss about existing governmental programs (if any). If there is no such program runs in the country, it can be recommended to the government to initiate such programs for this deprived section of the society.

6. PLOS authors have the option to publish the peer review history of their article (what does this mean?). If published, this will include your full peer review and any attached files.

Reviewer #1: Yes: S M Yasir Arafat

Reviewer #2: Yes: Sujita Kumar Kar

---

## [Author Response · Author response to Decision Letter 0]

5 Jun 2020

May 21, 2020

Dr. Russell Kabir, PhD

Academic Editor

PLOS ONE

Ms. Ref. No.: PONE-D-20-02224

Title: SUICIDAL BEHAVIOR RISKS DURING ADOLESCENT PREGNANCY IN A LOW-RESOURCE SETTING: A QUALITATIVE STUDY

Dear Dr. Kabir,

We are pleased to resubmit the above referenced manuscript for consideration for publication in PLOS ONE. We appreciate the time and effort of the reviewers, their recognition of the significance of our study, and their notation of several strengths of our manuscript and consideration for revision. We have addressed each concern detailing the sections where revisions are made using track changes within the manuscript. The revisions are outlined below.

Editor comments

We have taken note of this and made appropriate changes in the revised manuscript.

Please ensure that you refer to Figure 1 in your text as, if accepted, production will need this reference to link the reader to the figure.

Thank you for highlighting this. We have made reference to figure 1 in the text under methods: theoretical framework (page 5) 

REVIEWER REPORTS:

Reviewer #1: ADOLESCENT PREGNANCY IN A LOW-RESOURCE SETTING is an important issue related with suicidal behavior, however understudied. I thank authors for their important work in a country setting like Kenya which could provide insights to build the national suicide prevention strategies. Thanks.

Thank you for your time and positive feedback on our manuscript.

Reviewer #2: 

1. Refer to page 11 (Introduction section); Rephrase the sentence “We aimed to investigate the suicidal behaviour risks of adolescent pregnant mothers to inform the development of suicidal risk reduction for early detection at the community level and to attenuate maternal and child health complications post birth.”. It is too complex. It may be broken to two or three sentences.

Thank you for raising this, we have now restructured and simplified this sentence on page 5 (introduction), lines 4-7 to read “We aimed to investigate the suicidal behaviour risks of adolescent pregnant mothers. We believe that such a study would inform the development of suicidal risk reduction strategies with a focus on early detection at the community level. Additionally, implementation of the strategies would attenuate maternal and child health complications post birth.”

2. What was the selection criteria for choosing the participants for FGDs?

We have provided additional information for our selection criteria on page 6 under methods: Sampling and Data collection, paragraph 2 as follows: 

In our selection criteria, all participants needed to have resided in the study area for at least six months and willing to participate in the study voluntarily. The age requirement for the adolescent participants (either pregnant or below 6 weeks post natal) ranged between 13 to 19 years. The criterion for the TBAs was being ‘active’ with community activities. This was reflected in the monthly referrals of ante natal and post natal mothers to a primary health care facility. The public health officer in the study areas had the records of the TBA activities.

3. Whether the authors also included adolescent mothers, who became pregnant after non-consensual sex / rape?

Thank you for highlighting this. We have added information on page 8 (results) lines 15-16 to indicate that there was no adolescent pregnancy resulting from rape.

4. The authors need to mention the limitations of this study and recommendations to overcome limitations.

Thank you for this suggestion. We have added the following information on page 16-17 (Study limitation)

This study was qualitative and therefore context specific. A mixed methods study could augment study findings and draw general conclusions. In terms of the study area, this research was conducted in rural settings; therefore the findings might not reflect the actual experience in urban settings. Future studies could explore suicidal behavior in the context of urban pregnant adolescent population and make informed strategic comparisons in both areas.

5. In the implications, the authors may discuss about existing governmental programs (if any). If there is no such program runs in the country, it can be recommended to the government to initiate such programs for this deprived section of the society.

Thank you for this suggestion, we have now provided details on future implications and plans on page 16 (implications):

Lack of elaborate support systems for the adolescent perinatal mothers makes them vulnerable to suicidal behaviour. The paucity of mental health well-being programs in the rural areas makes the situation dire. It is plausible that the government initiated youth friendly clinics can mitigate the challenges which are specific to the youth. However, the clinics are mainly found in the urban referral hospitals. We recommend that youth-tailored services should be decentralized to the rural areas and especially inclusion of mental health services for adolescents.

---

## [Decision Letter · Decision Letter 1]

6 Jul 2020

SUICIDAL BEHAVIOR RISKS DURING ADOLESCENT PREGNANCY IN A LOW-RESOURCE SETTING: A QUALITATIVE STUDY

PONE-D-20-02224R1

Dear Dr. Musyimi,

We’re pleased to inform you that your manuscript has been judged scientifically suitable for publication and will be formally accepted for publication once it meets all outstanding technical requirements.

Kind regards,

Frank T. Spradley

Academic Editor

PLOS ONE

Reviewers' comments:

Reviewer's Responses to Questions

**Comments to the Author**

1. If the authors have adequately addressed your comments raised in a previous round of review and you feel that this manuscript is now acceptable for publication, you may indicate that here to bypass the “Comments to the Author” section, enter your conflict of interest statement in the “Confidential to Editor” section, and submit your "Accept" recommendation.

Reviewer #1: (No Response)

Reviewer #2: All comments have been addressed

2. Is the manuscript technically sound, and do the data support the conclusions?

Reviewer #1: (No Response)

Reviewer #2: Yes

3. Has the statistical analysis been performed appropriately and rigorously? 

Reviewer #1: (No Response)

Reviewer #2: Yes

4. Have the authors made all data underlying the findings in their manuscript fully available?

Reviewer #1: (No Response)

Reviewer #2: Yes

5. Is the manuscript presented in an intelligible fashion and written in standard English?

Reviewer #1: (No Response)

Reviewer #2: Yes

6. Review Comments to the Author

Reviewer #1: I thank authors for the modification and submitting the revision where comments have been addressed.

Reviewer #2: Revision is satisfactory. The authors have addressed to all the queries raised. It will be an interesting peace of publication with respect to the LMICs.

7. PLOS authors have the option to publish the peer review history of their article (what does this mean?). If published, this will include your full peer review and any attached files.

Reviewer #1: **Yes: **S M Yasir Arafat

Reviewer #2: No

---

## [Editor Report · Acceptance letter]

10 Jul 2020

PONE-D-20-02224R1 

Suicidal behavior risks during adolescent pregnancy in a low-resource setting: a qualitative study 

Dear Dr. Musyimi:

I'm pleased to inform you that your manuscript has been deemed suitable for publication in PLOS ONE. Congratulations! Your manuscript is now with our production department. 

Kind regards, 

on behalf of

Dr. Frank T. Spradley 

Academic Editor

PLOS ONE